# Deciphering Multifactorial Correlations of COVID-19 Incidence and Mortality in the Brazilian Amazon Basin

**DOI:** 10.3390/ijerph19031153

**Published:** 2022-01-20

**Authors:** Blanca Elena Guerrero Daboin, Italla Maria Pinheiro Bezerra, Tassiane Cristina Morais, Isabella Portugal, Jorge de Oliveira Echeimberg, André Evaristo Marcondes Cesar, Matheus Paiva Emidio Cavalcanti, Lucas Cauê Jacintho, Rodrigo Daminello Raimundo, Khalifa Elmusharaf, Carlos Eduardo Siqueira, Luiz Carlos de Abreu

**Affiliations:** 1School of Medicine, University of Limerick, V94 T9PX Limerick, Ireland; bgdaboin@yahoo.com (B.E.G.D.); andreemcesar@uol.com.br (A.E.M.C.); mpaivaemidio@gmail.com (M.P.E.C.); khalifa.elmusharaf@ul.ie (K.E.); carlos.siqueira@umb.edu (C.E.S.); 2School of Sciences of Santa Casa de Misericórdia de Vitória (EMESCAM), Vitoria 29045-402, Brazil; italla.bezerra@emescam.br (I.M.P.B.); tassiane.morais@emescam.br (T.C.M.); 3Department of Integrated Health Education, Federal University of Espirito Santo, Vitoria 29075-910, Brazil; 4Department of Internal Medicine, School of Medicine, University of Sao Paulo, São Paulo 05403-000, Brazil; iportugal@usp.br; 5Laboratory of Studies Design and Scientific Writing, Postgraduate Division, University Center FMABC, Santo André 09060-870, Brazil; bioredebrasil@gmail.com (J.d.O.E.); rodrigo.raimundo@fmabc.br (R.D.R.); 6Division of Immunology and Allergy, Department of Medicine, School of Medicine, University of Sao Paulo, São Paulo 05403-000, Brazil; lucas.caue@fm.usp.br; 7Department of Urban Planning and Community Development, School for the Environment, University of Massachusetts Boston, Boston, MA 02125, USA

**Keywords:** Amazonas, COVID-19, incidence, lethality, mortality, trends

## Abstract

Amazonas suffered greatly during the COVID-19 pandemic. The mortality and fatality rates soared and scarcity of oxygen and healthcare supplies led the health system and funerary services to collapse. Thus, we analyzed the trends of incidence, mortality, and lethality indicators of COVID-19 and the dynamics of their main determinants in the state of Amazonas from March 2020 to June 2021. This is a time-series ecological study. We calculated the lethality, mortality, and incidence rates with official and public data from the Health Department. We used the Prais–Winsten regression and trends were classified as stationary, increasing, or decreasing. The effective reproduction number (Rt) was also estimated. Differences were considered significant when *p* < 0.05. We extracted 396,772 cases of and 13,420 deaths from COVID-19; 66% of deaths were in people aged over 60; 57% were men. Cardiovascular diseases were the most common comorbidity (28.84%), followed by diabetes (25.35%). Rural areas reported 53% of the total cases and 31% of the total deaths. The impact of COVID-19 in the Amazon is not limited to the direct effects of the pandemic itself; it may present characteristics of a syndemic due to the interaction of COVID-19 with pre-existing illnesses, endemic diseases, and social vulnerabilities.

## 1. Introduction

The COVID-19 pandemic has heavily struck Brazil; it has the second highest number of deaths, just behind the United States of America [1]. Since the pandemic began, the circumstances in Brazil have been deeply marked by controversies over the government’s response [2]. Actions and implementation of distancing strategies vary from region to region and depend essentially on state governments [3].

Of all 27 Brazilian federative units, Amazonas has suffered the most from the pandemic [4]. After its first SARS-CoV-2 case was reported in Manaus, its capital, on 13 March 2020, mortality and fatality rates soared. Manaus has faulty health infrastructure, 293 hospital beds between public and private, and eight ambulances [5]. The lack of oxygen and healthcare supplies led to the deaths of at least 40 patients, not to mention oxygen delivery to remote areas representing another big challenge. Both the health system and funerary services collapsed and as a result, mass graves were dug [6]. These events caused emotional, personal, religious, social, legal, and economic distress for the families of those buried there and the survivors, significantly impacting local communities’ mental health and quality of life.

Several factors, including mobility, led to the COVID-19 crisis in Amazonas [7]. In this remote area, the international airport of Manaus has a flow of around three million passengers between arrivals and departures per year [8]. This volume of arrivals is mostly due to the Manaus Free Zone, a tax-reduced area, which has attracted several national and international industries and buyers [9].

Due to its geographical characteristics, ecosystems, flora, and fauna, Amazonas is a national and international tourist attraction for the practice of ecotourism and sportfishing [10]. A recurrent migratory flow from Venezuela, Colombia, and Peru comes through its borders, especially Venezuela. High fluvial mobilization is a characteristic of this region; communities from more remote areas move through its rivers to more urban areas, including indigenous native populations. The fluvial transit is not limited to nationals settled in upper river towns and other Brazilian neighborhoods; the port of Manaus is considered the largest floating one globally; tourist cruises and cargo ships arrive there [11].

Amazonas contains the country’s largest group of indigenous people [12]. According to the latest population census, the indigenous population in Amazonas is made up of around 168,000 individuals who are grouped into 62 communities [13].

These heterogeneous people flows come together in a territory labeled one of the most impoverished Brazilian areas [4] with significant income inequality [14]. These existing socioeconomic discrepancies impacted the course of the pandemic, determining poorer outcomes in municipalities with highly vulnerable populations [15], mainly in rural areas.

Specific features of the geographic area where an individual or a community belongs significantly influence the health–disease process. This fact is particularly relevant in the Amazon region due to its abundant water reservoir and extensive green areas. For instance, the state contains 20% of Earth’s freshwater reserves and around 31% of the planet’s tropical forests [16]. However, the urban and rural convergence faces insufficient water and sanitation systems, scarce health resources, and the government’s delayed response [17]. Furthermore, the rise in deforestation and the consequent dryness can complicate the COVID-19 pandemic and put the population residing in the Amazon at risk [18].

Fires in the Amazonian territory are responsible for 80% of the environmental pollution caused by the increase in particulate matter (PM2.5). To Johns Hopkins University [19], the possible relation between contact with PM2.5 and COVID-19 has special significance for public health in Brazil, where incidence and mortality rates are among the highest worldwide. People settled in the North Brazilian region, especially vulnerable communities, are particularly affected by high exposure; for instance, the mortality rate from COVID-19 among indigenous people is above the Brazilian average [20]. Moreover, there is evidence that the reported number of COVID-19 victims among the indigenous population are underestimated [21]. In addition, official data from the Brazilian Ministry of Health show that young adults’ mortality increased significantly in the first semester of 2021 compared to the same period of 2020 [22].

Several lineages with a temporal prevalence have been identified in Amazonas. Naveca et al.’s 2021 findings [23] reinforce the idea that consecutive lineage replacements were compelled by a complex mixture of varying grades of social distancing actions and the emergence of a more transmissible virus variant.

We are aware of the importance of monitoring the pandemic in the state of Amazonas, whose number of deaths exceeded 13,000 victims by the end of the first semester of 2021. This number is above what several countries in America and other continents have reported [24]. However, several COVID-19 studies on Amazonas focus on the characterization of individuals at a particular time [25]. The analysis and understanding of the temporal variations of COVID-19 mortality and lethality can lead to opportunely and satisfactory interventions for the prevention of more harmful consequences [26,27,28,29].

Thus, this study aimed to analyze the incidence, mortality, and lethality indicators of COVID-19 and the dynamics of their main determinants from March 2020 to June 2021.

## 2. Materials and Methods

### 2.1. Study Design and Location

Based on the protocol by Abreu, Emulsharaf, and Siqueira [30] for ecological time-series studies, we extracted public and official secondary data on cases of and deaths from COVID-19 from the dashboard of the Amazonas State Health Department, Brazil [31].

The state comprises an area of more than 1.5 million km^2^, 18% of the country’s territory, the most extensive of the Brazilian federative units with the lowest population density in the country (2.2 inhabitants/km^2^). However, Manaus, the central urban city and metropolitan area, has around 2 million inhabitants [14].

### 2.2. Sampling and Eligibility Criteria

We included all cases of and deaths from COVID-19 from March 2020 to June 2021. The occurrences were confirmed by laboratory diagnosis; clinical and clinical epidemiological COVID-19 were categorized according to the International Classification of Diseases, 10th edition (ICD-10) as “U07.1 COVID-19, identified virus” or “U07.2 COVID-19, virus not identified”, respectively [32].

According to the last day of care, we classified the deaths and cases by symptom onset date. Then, a second author verified the extracted data, and a third researcher made a final check in case of discrepancies. Lastly, the information was written in an Excel spreadsheet (Microsoft Corporation, Redmond, WA, USA) to further analyze the effective reproductive number (Rt), case fatality, mortality, and incidence.

### 2.3. Statistical Analysis

The incidence and mortality rates per 100,000 inhabitants and the case fatality (%) were determined with Equations (1)–(3). The estimations were based on the number of Amazon state residents in 2020 (4,240,210 inhabitants) [33]:(1)$$\mathrm{Incidence}=\frac{\mathrm{number}\;\mathrm{of}\;\mathrm{cases}}{\mathrm{population}}\times 100,000$$
(2)$$\mathrm{Mortality}=\frac{\mathrm{number}\;\mathrm{of}\;\mathrm{deaths}}{\mathrm{population}}\times 100,000$$
(3)$$\mathrm{Case}\;\mathrm{fatality}=\frac{\mathrm{number}\;\mathrm{of}\;\mathrm{deaths}}{\mathrm{number}\;\mathrm{of}\;\mathrm{cases}}\times 100$$

For trend analysis, the period was divided into two waves: 1st—March to September 2020 and 2nd—October 2020 to June 2021. The trend analysis followed the protocol of Antunes and Cardoso [34]. The time-series were built applying the Prais–Winsten regression model for population mortality rates. It allowed the first-order autocorrelation to analyze time series values and facilitate the assessment and classification of mortality and case fatality into increased, decreased, or flat. Trends were classified as flat when the *p*-value was not significant (*p* > 0.05).

The values for probability (*p*) and daily percent change (DPC), considering a 95% level of significance, were calculated using Equations (4)–(6), where *β* is the angular coefficient from the linear regression, the indexes *ul* mean the upper limit, and *ll* is the lower limit of the confidence level.
(4)$$DPC=\left(10^{\beta }-1\right)\times 100\%\;$$
(5)$${\left(CI95\%\right)}_{ul}=\left(10^{\beta_{max}}-1\right)\times 100\%$$
(6)$${\left(CI95\%\right)}_{ll}=\left(10^{\beta_{min}}-1\right)\times 100\%$$

To compare proportions, the two-tailed z-test was used, considering the differences with *p*-value < 0.05 as significant.

Statistical analyses were performed using the STATA 14.0 software (College Station, TX, USA, 2013). The Rt was estimated using R studio software EpiEstim package [35] version 2.2.4, a time-varying reproduction number for epidemics developed by Thompson and colleagues [36]. As described in previous studies, we used a mean serial interval of 2.97 days with a mean standard deviation of 3.29 days [37,38]. Then, we performed a Spearman correlation analysis between Rt and daily new cases, Rt and daily deaths, and Rt and lethality.

## 3. Results

The data indicated 396,772 cases of and 13,420 deaths from COVID-19 during the examined period across different municipalities.

The impact of COVID-19 cases and deaths in the Brazilian Amazon was visible throughout its territory. The number of fatal victims was significantly higher in urban areas, with Manaus accounting for 47% of cases and 69% of deaths. The municipalities of Itacoatiara and Iranduba, located near Manaus, registered 17,328 cases and 527 deaths between them (Table 1). However, the virus also reached remote areas where indigenous communities are settled, such as the municipalities of San Gabriel da Cachoeira, Labrea, Antazes, Borba, Barcelos, Atalaia do Norte, and Manicore. Figure 1 illustrates the geographical location of these distant lands that reported 28,059 cases and 524 deaths from COVID-19 and Table 1 indicates the distribution of confirmed cases and deaths by municipality.

On characterizing the infected population in the 62 municipalities of the state of Amazonas, and later by tabulating the data, we verified that age or sex information was missing in 80 cases (0.02%) reported by the health secretaries. Therefore, calculations concerning the variables of age and sex were made based on 396,692 cases. Figure 2 splits into graphs A and B and provides details of the distribution by age and sex of the population affected by COVID-19.

Concerning the number of cases (Figure 2a), the virus affected the female population more, with 220,320 cases (56% with *p*-value < 0.001). The z-test for proportions made it possible to compare the proportions in the number of cases of infected people and deaths by sex and age group.

The proportions of cases were higher in females in the 20-to-29-year-old group (*p*-value < 0.001) and 30 to 39 years (*p*-value < 0.001). They were equal in the age groups 0 to 19 years (*p*-value = 0.565) and 40-to-49-year-old (*p*-value = 0.204). For the age group 50 years old or more, the proportion of male cases was higher (*p*-value < 0.001)

Figure 2b indicates that 66% (*p*-value < 0.001) of deaths occurred in the population over 60 years of age; 57% were men; there was no significant difference between the proportions of deaths in the two sexes because the lowest *p*-value was 0.162 in this age group. The group aged 50 to 59 accounted for 16% of deaths, and 60% were men (*p*-value = 0.048). Young adults between 20 and 49 years accounted for 6% of deaths and 59% were male (*p*-value < 0.001).

Through the z-test, it was possible to compare the confirmed cases group and the group of deaths from COVID-19 in the presence of comorbidities.

By analyzing fatal COVID-19 cases and related comorbidities, notice in Table 2 that cardiovascular diseases (CVD) were most commonly found among deaths with *p*-value < 0.001 with a proportion equal to 28.84%, followed by diabetes (25.35%) and other comorbidities (25.29%), respectively with *p*-value = 0.912.

On the other side, immune-depressed diseases (32.98%) were the most reported comorbidity (*p*-value < 0.001), followed by CVD (24.98% with *p*-value < 0.001) and diabetes (18.75% with *p*-value < 0.001).

The transmissibility of SARS-CoV-2 fluctuated over the period analyzed. The Rt reached its highest value in April 2020 during the first wave and the end of December 2020 during the second wave (Figure 3). The observed values are consistent with the peaks of accumulated cases and their impact on mortality observed in May 2020 and January 2021.

Figure 3 shows the Rt as a function of time for the study period. It showed significant growth between April and May 2020 and another rise in January 2021; for the other months, stationary behavior was observed. This pattern appears similar to the time series of new cases; therefore, correlation analyzes were performed between Rt and daily new cases, Rt and daily deaths, and between Rt and lethality. Table 3 illustrates the results for Spearman’s ρ coefficients.

As shown in Table 3, the correlation coefficient between Rt and deaths was not significant; that is, the behavior of the frequency of deaths is not associated with Rt, as expected (ρ ≅ 0). There is a positive and significant correlation between Rt and new cases. However, this correlation is not as strong as expected (ρ ≅ 1), with the coefficient ρ = 0.264. For the correlation between Rt and lethality, ρ = −0.334 was found, suggesting a significant negative association between Rt and lethality.

Our data revealed two well-defined periods in the pandemic. Table 4 indicates cases, deaths, case fatality (%), mortality, and incidence rates of COVID-19 per 100,000 inhabitants during the two waves. During the first period, a peak in mortality and incidence occurred in May 2020. Then, the numbers remained stable until September 2020. However, January 2021 was characterized by an exponential increase in cases where the incidence reached its highest number per 100,000 inhabitants. Almost six thousand fatalities were reported between January and February of 2021; the mortality rate peaked in January 2021 (85.84 deaths per 100,000 inhabitants).

Table 5 displays the analyses of incidence, lethality, and mortality trends. During the first wave, incidence rates registered increasing trends, with a DPC of 0.62%, while lethality showed decreasing trends, recording a DPC of −0.50% (*p* < 0.05). The incidence trend fell during the second wave with a DPC of −0.42%, a significant difference. The mortality trends were flat for both waves, with a non-significant *p*-value (*p* > 0.05).

## 4. Discussion

The ability to examine the spread rate, mortality, fatality, and factors that affect these indicators is essential to understanding the trends of the COVID-19 pandemic. We analyzed the scenario from demographic, geographical, and socioeconomic perspectives.

The projection for the population [33] indicates that 57% are under 30 years old. The age composition reflects an expansive population pyramid shape in which 38% are aged 0–19 years, 19% are 20–29, 15.8% are 30–39, 12% are 40–49, 7.9% are 50–59, and 4.5% are 60–69 years. Less than 3% of the population is 70 years or older.

During the examined period, Amazonas faced two critical episodes, the first in May 2020, reaching 1615 deaths and 36,123 new cases. The second even worse critical peak was in January 2021, when the number of deaths and number of cases were almost double those of May 2020, which led to a collapse of the local health system. These findings coincide with published evidence [23].

From May to the first week of December 2020, fatalities decreased to values very similar to the Brazil national average, varying between 2% and 3%. During those seven months, the hospitalizations in Manaus had low variation [39], influenced by an increase in the number of beds and the implementation of better protocols in primary care and the experience gained by health professionals to attend the most severe cases [40].

Then, politicians, health authorities, and the population relaxed prevention measures during Christmas because the population was “relatively protected”. Moreover, the findings of Buss et al. [6] indicate that 70% of the Manaus population had already been infected with COVID-19, which theoretically would imply herd immunity.

However, according to Ferrante et al. [41], Manaus had a confirmed case of reinfection and immunity loss with SARS-CoV-2 with the same virus lineage. In July 2020, a patient tested positive and the symptoms of reinfection with an aggravated condition were observed in October. Even with severe disease, IgG antibodies were not present in this patient, suggesting that he did not acquire natural immunity to SARS-CoV-2, minimizing the expectations of herd immunity both during the first infection and reinfection.

The accelerated increment of confirmed cases by COVID-19 began by the end of December 2020. It generated a critical effect on the number of fatalities in January 2021, influenced by a new, much more aggressive variant, with P1 being the most predominant.

A sequence of variants has marked the state of Amazonas; a genomic survey conducted from 16 March 2020, to 13 January 2021, revealed several lineages with a temporal prevalence in 25 municipalities of Amazonas, Brazil [23]. The survey reported that the lineage B.1.195 was the most prevalent during the peak experienced in late April and early May during the first wave. Its presence declined, and by the end of May, it was replaced by B.1.1.28, which prevailed until December 2020. Those mutations may allow the coronavirus to disseminate quickly by contact from person to person, and more infections can cause more individuals to get sick or die.

Galvão et al. [42] previously mentioned that the crisis experienced in Manaus and its nearby cities with the COVID-19 pandemic might be a consequence of this region’s limited health structure and social vulnerability. However, high mortality rates have been reported in cities of high-income countries [42]. Thus, this may suggest the existence of crucial determinants in COVID-19 morbimortality that are specific to the Brazilian Amazon, such as air pollution, fire seasons, and deforestation.

### 4.1. Quality of Air and Deforestation

Exposure to smoke from forest fires in the Amazon impacts the health of populations most at risk, including those with chronic heart or lung diseases, older adults, children, pregnant women, and fetuses [43]. Remarkably, air pollution has been proven to significantly impact morbidity and mortality from cardiovascular outcomes [44], mainly ischemic heart disease and stroke, which are the most prevalent comorbidity among COVID-19 deaths worldwide and in Amazonas, as mentioned in our results.

A study from the Imperial college and international collaborators [45] found that 76% of the Manaus population was infected by SARS-CoV-2 between March and October 2020, while in São Paulo, the percentage of people infected was 29%. The new, more contagious variant and the alert of reinfections put extreme pressure on the health system. The variant, known as P.1 or VOC202101/02 in the U.K., was first detected in January 2021 in Japan in people who visited Manaus. This variant has several mutations, including E484K and N501Y. It involves 17 unique amino acid changes, three deletions, four synonymous (silent) mutations, and one 4nt insertion.

The interaction between weather, nature, and people’s actions is a crucial determinant in Amazonian fires. Human-induced fires have been occurring in the Brazilian Amazon since the late 1980s [46,47]. Most of them happen between July and November, during the dry period, rising in September for the south of the basin [48]. Dryness grows the fire risk due to reduced groundwater and soil moisture [49]. Utilizing water surface temperature in the Atlantic and Pacific sea, statistical models predict fire activity [50]. Nevertheless, such prognostications based only on climate miss the vital function of people across the Amazon.

Local populations have been consistently exposed to PM2.5 for over 30 years, characterizing long-term harmful exposure to this air pollutant as observed in previous studies in the Amazon [51] and Equatorial Asia [52,53]. Our findings reveal that most of the fatal victims (66%) were people older than 60 years; most of them (57%) were males. Consistently, research among the Brazilian regions reported that mortality in Manaus in people older than 70 years was double that in Rio de Janeiro and three times that of São Paulo. The most common comorbidities were CVDs and diabetes, similar to those reported in other studies worldwide [54,55].

Freitas et al. [56] observed an increase in the incidence of COVID-19 among younger adults during the second wave (January 2021). We identified that six percent of deaths occurred in adults between 20 and 49 years; 56% of the male sex. In this study, we did not identify the risk factors by age group. Nonetheless, research from the FVS-AM (Amazon Health Surveillance Foundation) during the first quarter of 2021 reported obesity as the main comorbidity among people who develop severe COVID-19 and die in the age range of 20–39 years [57].

In addition, the emergence of long-term effects among people who have suffered from COVID-19 has been documented. A systematic review reported more than 50 conditions in post-COVID-19 patients [58]. This suggests that the high mortality [59] and COVID-19 sequelae can potentially decline life expectancy.

Our analysis indicates that the scenario changed from an incidence-increasing trend during the first wave to a decrease during the second one. The growing evolution of first wave incidence remained constant until March 2021. The growth in the number of cases was deeply affected by the lack of non-pharmacological measures, such as social distancing [60] and the use of face masks. On the other hand, particularly in Brazil, the lack of a national clinical guideline makes it challenging to evaluate the reasons for high mortality from COVID-19, mainly in intubated patients.

The incidence of COVID-19 in Amazonas was marked by a decreasing trend at the end of June 2021, which suggests the positive effect of the vaccination program. It began in mid-January 2021. Older adults, people with disabilities, health professionals, and the indigenous population were within the priority group [61]. By the end of February, more than 60% of indigenous people over 18 had received the first dose [62].

The trend in mortality was stable for both defined waves. On the other side, the lethality decreased in the first wave to an increasing trend in the second wave. In this scenario, the number of COVID-19 tests was focused on severe patients, which suggests that the percentage of deaths could be higher than that reported [63].

The reduction in deaths from March 2021 is associated with prevention measures and the closure of non-essential businesses, especially in Manaus and its surroundings, where more than 50% of the state’s population is concentrated. In addition, the state of Amazonas became a priority in the national vaccination program [61]. Research by Orellana et al. [64] reported an excess of 312% (95% CI 304–321) among 46,028 deaths from respiratory causes that occurred in eight cities from February 23 to August 08, 2020. Manaus recorded the highest excess of mortality, with 758% (95% CI 668–858), and São Paulo the lowest, with 174% (95% CI 164–183). The significant differences among the cities reinforce the regional inequalities in health access aggravated by the lack of a national clinical protocol and suggest a potential underreporting of deaths due to COVID-19.

The maximum level of transmission, represented by the number of cases and deaths, was first reached in the northern region [40]. Thus, the epidemiological situation of the Amazon is of great concern. Before the pandemic, the health coverage problems and insufficient hospital beds, intensive care beds, and ventilators were already evident public health issues in this territory. The pandemic was a fuel that more starkly evidenced those limitations; Manaus, the capital, is the only city with intensive care units (ICU). A study by the Brazilian Association of Orthomolecular Medicine on the evolution of patients treated with invasive mechanical ventilation due to COVID-19 indicated that mortality in the Amazon was 88.5%, while that of the south and southeast of the country was 76.8%. However, in some reference hospitals in São Paulo the mortality rate in intubated patients fluctuated from 30% to 45% [65].

### 4.2. Territorial Variables Affecting the COVID-19 Crisis in Amazonas State

The limitations of access to health care and the social disparities in Northern Brazil have been decisive in the complex situation of high incidence and high mortality in Amazonas. However, as pointed out by Fundação Oswaldo Cruz [40], the daunting scenario reveals a combination of factors that favored the spread of this virus (Figure 4) and essentially justified the high incidence and mortality found in this state.

First, the state of Amazonas shares borders with Colombia, Venezuela, and Peru. Brazil delayed closing its borders to control the spread of the virus. A significant population of migrants, mainly from Venezuela, works in the informal economy, mainly as street vendors in Manaus. They are a group that does not comply with quarantine and distancing rules.

Second, the largest indigenous population in Brazil is concentrated in this state. Additionally, other communities such as river dwellers and descendants of former slaves (quilombolas) are settled in distant areas on the banks of the rivers [66]. They are highly vulnerable and face constant threats from illegal mining and logging in the Amazon rainforest. Miners and loggers are potential carriers of the virus and can infect rural communities [2]. The paradoxical high mortality in these sparsely populated areas may be attributable to the deficient sanitation and communal use of water reservoirs, in addition to crowded boats making transport through the rivers benefit the spread of the virus [17].

According to the Brazil Indigenous People Articulation [67] the first death of an indigenous person was in Amazonas. Up to 30 April 2021, this state experienced 254 indigenous deaths—the highest in the country.

Third, the Manaus Free Zone has international companies and frequently receives people from abroad. As mentioned, variant P.1 was identified in Japan among travelers who had visited Manaus. Later, this variant was identified in patient samples collected in January 2021 [68].

Fourth, Amazonas is an endemic area for Malaria and Dengue, which have some symptoms similar to COVID-19. Additionally, there is evidence of co-infection of SARS-CoV-2 with dengue [69]. This syndemic context makes the detection and treatment of COVID-19 complex in the region.

This study may present limitations inherent to secondary databases, such as alterations in terms of residence, which can cause distortions in the numbers of cases and deaths by city or municipality.

The results presented are partial data since the pandemic is still being analyzed. The data were consolidated based on the last day’s report, but these may have been reported late. The number of accumulated cases may be higher, considering the limitations of doing massive tests to detect the disease. Hence, it is pertinent to mention potential discrepancies between the reported incidence and the actual number of people infected with the virus.

## 5. Conclusions

The impact on mortality and incidence by COVID-19 in the Amazon is not limited to harm generated by the pandemic. Various region-specific factors may have significantly contributed to the Brazilian Amazon’s complex scenario during the COVID-19 pandemic, such as poor sanitation systems, dependence on communal water reservoirs, and long-term exposure to air pollutants due to forest fires. Furthermore, it may present the characteristics of a syndemic due to the interaction of pre-existing diseases and social vulnerability.

Our results highlight the importance of systematic and persistent efforts to research and develop interventions seeking environmentally sustainable development of the Brazilian Amazon and improvements in healthcare quality and access to socioeconomically vulnerable populations. Hence, it is essential to carry out systematic population studies with pandemic data to support prevention and control strategies.

## Figures and Tables

**Figure 1 ijerph-19-01153-f001:**
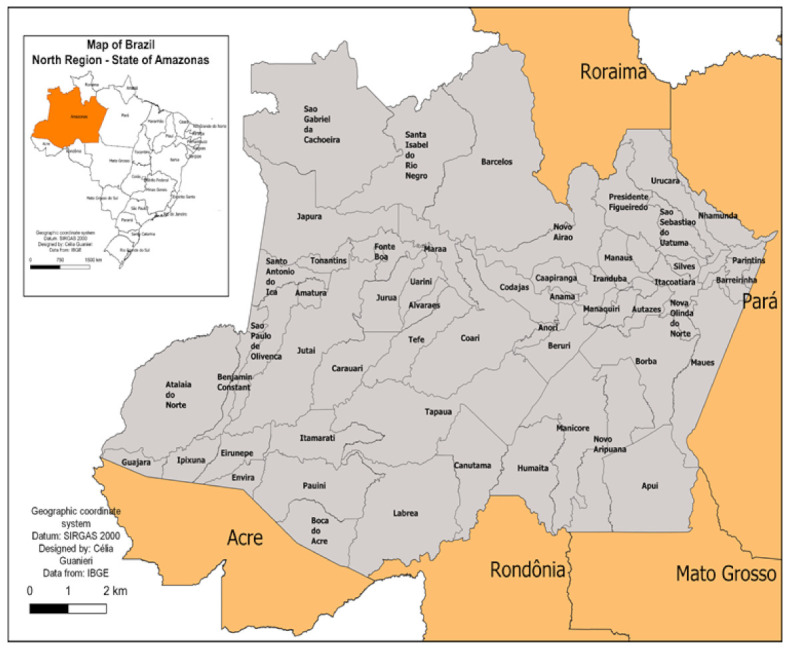
Map of the Amazonas state, indicating its municipalities.

**Figure 2 ijerph-19-01153-f002:**
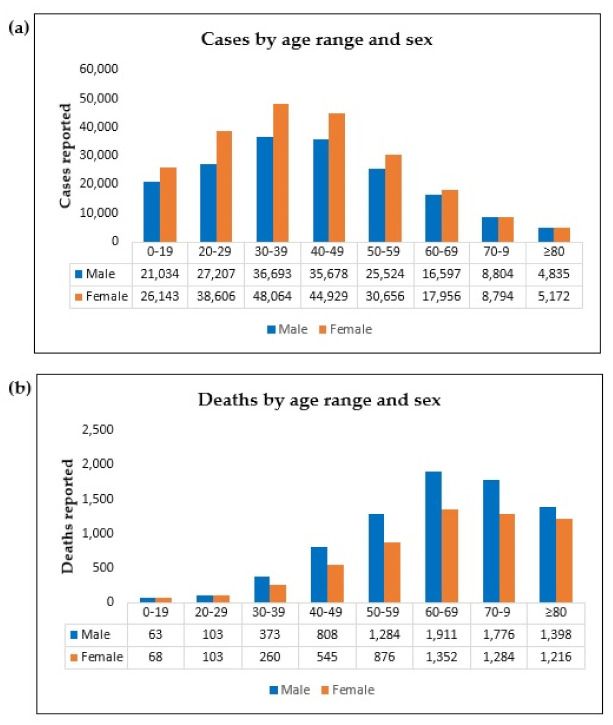
The number of COVID-19 cases (**a**) and deaths (**b**) reported by age range and sex in Amazonas State, March 2020–June 2021. Source: Cases and Deaths from the Secretary of Health of the Amazonas State. Note: For the number of cases analysis, 80 cases were excluded by missing information of age or sex.

**Figure 3 ijerph-19-01153-f003:**
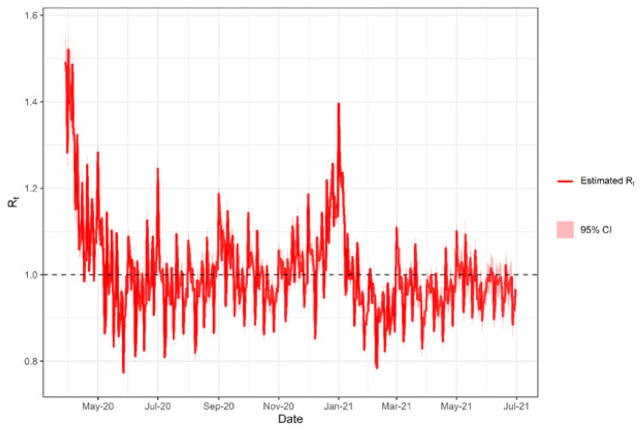
Effective reproduction number (Rt) estimated of COVID-19 from March 2020 to June 2021 in the state of Amazonas, Brazil. CI = confidence interval. Rt = effective reproduction number.

**Figure 4 ijerph-19-01153-f004:**
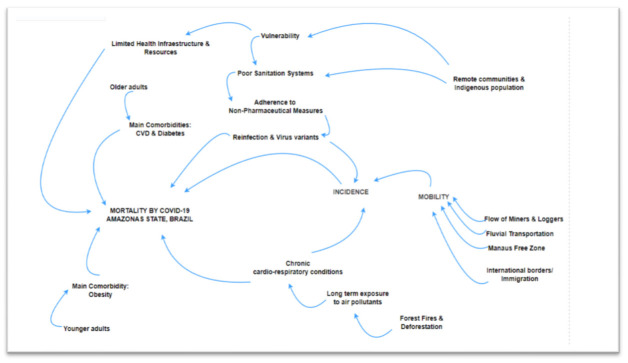
Cause consequence diagram of factors affecting mortality by COVID-19 in Amazonas State, March 2020–June 2021.

**Table 1 ijerph-19-01153-t001:** Confirmed cases and deaths by municipalities, Amazonas State, March 2020–June 2021.

Municipality	Cases		Deaths	
	*n*	%	*n*	%
Alvarães	2199	0.55	38	0.28
Amaturá	1227	0.31	17	0.13
Anamã	1443	0.36	9	0.07
Anori	1976	0.50	40	0.30
Apuí	1408	0.35	31	0.23
Atalaia do Norte	2737	0.69	13	0.10
Autazes	3366	0.85	99	0.74
Barcelos	3662	0.92	64	0.48
Barreirinha	2444	0.62	64	0.48
Benjamin Constant	3499	0.88	93	0.69
Beruri	1580	0.40	33	0.25
Boa vista do Ramos	1029	0.26	15	0.11
Boca do Acre	2708	0.68	24	0.18
Borba	2393	0.60	66	0.49
Caapiranga	618	0.16	21	0.16
Canutama	1035	0.26	10	0.07
Carauari	4979	1.25	54	0.40
Careiro	3949	1.00	84	0.63
Careiro da Várzea	2587	0.65	21	0.16
Coari	9400	2.37	226	1.68
Codajás	3006	0.76	21	0.16
Envira	3347	0.84	11	0.08
Eurinepé	3857	0.97	33	0.25
Fonte Boa	2726	0.69	35	0.26
Guajará	1577	0.40	25	0.19
Humaitá	7761	1.96	103	0.77
Ipixuna	4458	1.12	20	0.15
Iranduba	7510	1.89	166	1.24
Itacoatiara	9818	2.47	361	2.69
Itamarati	589	0.15	7	0.05
Itapiranga	3091	0.78	39	0.29
Japurá	1045	0.26	11	0.08
Juruá	1114	0.28	18	0.13
Jutaí	1843	0.46	28	0.21
Lábrea	3257	0.82	82	0.61
Manacapuru	12,447	3.14	398	2.97
Manaquiri	1582	0.40	44	0.33
Manaus	186,509	47.01	9186	68.45
Manicoré	4798	1.21	93	0.69
Maraã	2206	0.56	30	0.22
Maués	4508	1.14	136	1.01
Nhamundá	1821	0.46	33	0.25
Nova olinda do Norte	1598	0.40	70	0.52
Novo Airão	2552	0.64	28	0.21
Novo Aripuanã	1245	0.31	25	0.19
Parintins	15,176	3.82	327	2.44
Pauini	2298	0.58	22	0.16
Presidente Figueiredo	5879	1.48	113	0.84
Rio Preto da Eva	4254	1.07	80	0.60
Santa Isabel do Rio Negro	2262	0.57	46	0.34
Santo Antônio do Içá	2178	0.55	54	0.40
São Gabriel da Cachoeira	7846	1.98	107	0.80
São Paulo de Olivença	4095	1.03	71	0.53
São Sebastião do Uatumã	1074	0.27	20	0.15
Silves	1438	0.36	20	0.15
Tabatinga	2831	0.71	130	0.97
Tapauá	1506	0.38	11	0.08
Tefé	8670	2.19	245	1.83
Tonantins	1024	0.26	25	0.19
Uarini	1868	0.47	29	0.22
Urucará	2795	0.70	56	0.42
Urucurituba	3072	0.77	39	0.29
Without municipality identification	2	0.00	0	0.00
Total	396,772	100.00	13,420	100.00

**Table 2 ijerph-19-01153-t002:** Comorbidities reported by cases and deaths due to COVID-19, in Amazonas State, from March 2020 to June 2021.

Comorbidity	Cases		Deaths	
	*n*	%	*n*	%
Chronic obstructive pulmonary disease	5833	6.78	545	3.98
Cardiovascular disease	21,502	24.98	3948	28.84
Obesity	3887	4.52	768	5.61
Down’s syndrome	942	1.09	46	0.34
Hematologic disease	238	0.28	97	0.71
Immunodepressants	28,391	32.98	268	1.96
Neurological disease	716	0.83	396	2.89
Kidney disease	1462	1.70	565	4.13
Liver disease	236	0.27	125	0.91
Diabetes	16,140	18.75	3471	25.35
Other comorbidities	6735	7.82	3462	25.29
Total	86,082	100.00	13,691	100.00

**Table 3 ijerph-19-01153-t003:** Study of Spearman’s correlation between Rt and lethality, new cases, and deaths.

Analysis	ρ (CI 95%)	Freedom Grades	Statistics S	*p*-Value
R_t_ x case fatality	−0.334 (−0.412: −0.240)	457	21,503,589	<0.001
R_t_ x New cases	0.264 (0.170: 0.349)	457	11,864,418	<0.001
R_t_ x Deaths	−0.076 (−0.163: 0.016)	457	17,340,448	0.104

*p*-value: the probability of statistical significance. Source: The number of new cases and deaths were extracted from the Department of Health of State of Amazonas, Brazil.

**Table 4 ijerph-19-01153-t004:** The number of new cases, deaths, case fatality (%), mortality, and incidence rate by 100,000 inhabitants of COVID-19 in Amazonas State from March 2020 to June 2021.

Wave	Date	New Cases (*n*)	Deaths (*n*)	Case Fatality (%)	Mortality Rate (per 100,000 Inhabitants)	Incidence Rate (per 100,000 Inhabitants)
1st wave	March	3734	4	0.11	0.09	88.06
	April	26,121	1252	4.79	29.53	616.03
	May	38,583	1612	4.18	38.02	909.93
	June	27,075	522	1.93	12.31	638.53
	July	23,196	325	1.40	7.66	547.05
	August	17,290	265	1.53	6.25	407.76
	September	21,598	263	1.22	6.20	509.36
2nd wave	October	19,747	397	2.01	9.36	465.71
	November	18,456	341	1.85	8.04	435.26
	December	33,572	532	1.58	12.55	791.75
	January	75,037	3640	4.85	85.84	1769.65
	February	34,334	2254	6.56	53.16	809.72
	March	23,909	925	3.87	21.81	563.86
	April	13,620	530	3.89	12.50	321.21
	May	11,868	311	2.62	7.33	279.89
	June	8632	247	2.86	5.83	203.57
Total	March 2020 to June 2021	396,772	13,420	3.38	316.49	9357.37

*n* = 396,772 cases and 13,420 deaths due COVID-19. 1st wave: from March to September of 2020. 2nd wave: from October 2020 to June 2021. Source: Cases, deaths were extracted from the Secretary of Amazonas.

**Table 5 ijerph-19-01153-t005:** Prais–Winsten regression estimates and DPC of case fatality (%) mortality and incidence rate by 100,000 inhabitants of COVID-19 in the state of Amazonas, during the 1st wave (March to September 2020) and the 2nd wave (October 2020 to June 2021).

Period	DPC (CI 95%) Case Fatality	*p*	Fatality Trends	DPC (CI 95%) Mortality	*p*	Mortality Trend	DPC (IC 95%) Incidence	*p*	Incidence Trend
1st wave	−0.50 (−0.80; −0.20)	0.001	Decreasing	0.96 (−0.54; 2.48)	0.208	Flat	0.62 (0.12; 1.11)	0.014	Increasing
2nd wave	0.22 (0.03; 0.41)	0.022	Increasing	−0.20 (−0.81; 0.42)	0.534	Flat	−0.42 (−0.68; −0.17)	0.001	Decreasing

DPC—daily percent change (%); CI 95%—confidence interval 95%; *p*-value—the probability of statistical significance. Statistical difference detected by the Prais–Winsten regression test, *p* < 0.05. Source: Number of cases and deaths were extracted from the Secretary of Health of Amazonas, Brazil.

## Data Availability

Data were extracted from a population database in a COVID-19 dashboard, freely accessible on the Health Department of the State of Amazonas website: http://saude.am.gov.br/painel/corona/ (accesed on 19 July 2021).

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
