# Peer review of "Deciphering Multifactorial Correlations of COVID-19 Incidence and Mortality in the Brazilian Amazon Basin"

_ijerph, 2022, doi:10.3390/ijerph19031153_

Round 1

Reviewer 1 Report

The authors tried to discuss the Deciphering multifactorial correlations of COVID-19 incidence and mortality in the Brazilian Amazon basin by analysing disease incidence, mortality, and lethality indicators from March 28 2020 to June 2021. The manuscript need extensive English editing. In addition, the authors depend mainly on the public or published data whis are already available however they did not try to do any additional work to show the novelity of this study. All analyses tools are not enough and extensive statistical analyses are needed.

Author Response

Dear Reviewer :

On behalf of the authors thanks for the feedback received on our manuscript entitled: Deciphering multifactorial correlations of COVID-19 incidence and mortality in the Brazilian Amazon Basin.

Based on your valuable comments, we have made some actions and changes, which we respectfully detail below. Please see the attachment.

Sincerely,

The authors.

Reviewer 2 Report

The authors of the manuscript entitled: Deciphering multifactorial correlations of COVID-19 incidence and mortality in the Brazilian Amazon basin, present the results of the correlation of COVID-19 with different factors, using a database of cases and confirmed deaths in 62 municipalities of the state of Amazonas in the period from March 2020 to June 2021.

The results obtained are like studies carried out in other states in Brazil and other countries.

The manuscript is of great importance because it shows us the correlation between COVID-19 and comorbidities and reflects the multiple health deficiencies in developing countries.

In general, it is a good contribution to the scientific community. However, multiple errors are presented in the manuscript writing, tables, and figures. It is necessary modify the following typos/mistakes:

Lane 27, remove the space between the comma and from

Lane 31, Should be “the effective reproduction number (Rt)”

Homogenize text, 

Example: Use of space or not space between a number, and percentage (%), Lanes 33,34, 35, 309, 310, 311, 317, 327, 328, 329, 337, 439

Also, please use a dot or comma between the number in all manuscripts and special in the indicated lanes.

Figure 1. It is difficult to read the names of the municipalities; perhaps the font style and letter size should be changed.

Lane 184: change Deaths for deaths

Lane 189: I am afraid that 80 cases of 396,770 is 0.02% and not 0.0002%

Lane 190: Also, I think the number should be 396,690 and not 396,692 (because 396,770-80cases=396,690).

Figure 2. Please homogenize the graphs, a has lanes, and b does not on the y axis.

Lane 231. Do not use a capital letter for Cases and Deaths.

Lane 233. Please check if the number of cases is correct (N=396,692 or 396,690?)

Table 3. Do not use capital letters for New and Deaths.

Lane 284. Do not use capital letters for New and Deaths.

Lane 304. Please check the N. Here, the authors said the N is 396,772 cases.

Table 5. Homogenize wave, Wave.

Lane 353. Could the authors explain the importance of SARS-CoV-2 variants? Moreover, which variants were identified using genomic tools?

Lane 379. Change Human by human 

Figure 4. Change Consequence by consequence 

Based on the incongruous numbers in the text, figures, and tables, I suggested that all the values described in the manuscript should be reviewed, perform a new statistical analysis with the correct numbers.

Author Response

(The authors gave the same response as above.)

Reviewer 3 Report

The quality of this paper is good, I recommend accepting it.

Actually, the results of this study are not attractive because they are mostly consistent with previous studies. However, I still recommend this study because, it is conducted in the Amazon basin, which has suffered a lot from the COVID-19 pandemic. Most previous studies are conducted in the US, Europe, or Asia, the Amazon basin area, however, has not attracted enough attention. Therefore, this study is an effective complement to the existing research. Another important reason I recommend this study is its good quality. They collected data from March 2020 to June 2021, which is long enough to observe the development process of the COVID-19 pandemic. Methods used in this study, such as Prais-Winsten regression are reasonable. The quality of presentation, the English language and style are also fine.

I have read this paper carefully, and my general impression is that it is good and meets the standard of the IJERPH journal.

Author Response

(The authors gave the same response as above.)

Round 2

Reviewer 1 Report

Thanks for responding to my comments, good luck

Reviewer 2 Report

Dear Authors,

I appreciate that you have made the changes to the manuscript. The data presented in this manuscript represents a good contribution to the scientific community.